# Experiences of people taking opioid medication for chronic non-malignant pain: a qualitative evidence synthesis using meta-ethnography

Vivien P Nichols ,[1,2] Francine Toye,[3] Sam Eldabe,[4] Harbinder Kaur Sandhu,[1] Martin Underwood,[1] Kate Seers[2]

¹Warwick Clinical Trials Unit, Warwick Medical School, University of Warwick, Coventry, UK
²Warwick Research in Nursing, Division of Health Sciences, Warwick Medical School, University of Warwick, Coventry, UK
³Physiotherapy Research Unit, Nuffield Orthopaedic Centre, Oxford University Hospitals NHS Trust, Oxford, UK
⁴Department of Pain Medicine, The James Cook University Hospital, Middlesbrough, UK

**Correspondence to**
Mrs Vivien P Nichols;
V.P.Nichols@warwick.ac.uk

## ABSTRACT

**Objective** To review qualitative studies on the experience of taking opioid medication for chronic non-malignant pain (CNMP) or coming off them.

**Design** This is a qualitative evidence synthesis using a seven-step approach from the methods of meta-ethnography.

**Data sources and eligibility criteria** We searched selected databases—Medline, Embase, AMED, Cumulative Index to Nursing and Allied Health Literature, PsycINFO, Web of Science and Scopus (Science Citation Index and Social Science Citation Index)—for qualitative studies which provide patients' views of taking opioid medication for CNMP or of coming off them (June 2017, updated September 2018).

**Data extraction and synthesis** Papers were quality appraised using the Critical Appraisal Skills Programme tool, and the GRADE-CERQual (Grading of Recommendations Assessment, Development and Evaluation working group - Confidence in Evidence from Reviews of Qualitative research) guidelines were applied. We identified concepts and iteratively abstracted these concepts into a line of argument.

**Results** We screened 2994 unique citations and checked 153 full texts, and 31 met our review criteria. We identified five themes: (1) reluctant users with little choice; (2) understanding opioids: the good and the bad; (3) a therapeutic alliance: not always on the same page; (4) stigma: feeling scared and secretive but needing support; and (5) the challenge of tapering or withdrawal. A new overarching theme of 'constantly balancing' emerged from the data.

**Conclusions** People taking opioids were constantly balancing tensions, not always wanting to take opioids, and weighing the pros and cons of opioids but feeling they had no choice because of the pain. They frequently felt stigmatised, were not always 'on the same page' as their healthcare professional and felt changes in opioid use were often challenging.

**Trial registration number** 49470934; Pre-results.

## Strengths and limitations of this study

► To our knowledge this is the first qualitative evidence synthesis of patients' experiences of taking opioid medications.
► Meta-ethnography provides a thorough, systematic way of synthesising qualitative findings across multiple studies.
► Meta-ethnography provides the reviewer's interpretation of second-order concepts.
► Using a GRADE-CERQual (Grading of Recommendations Assessment, Development and Evaluation working group - Confidence in Evidence from Reviews of Qualitative research) approach can assist in rating confidence in the review findings.
► Qualitative research that illuminates patients' perspectives can help to shape future approaches to opioid management.

can impact heavily on people's quality of life.[1 2] Opioid medications are strong painkillers which have a well-established role in the treatment of acute and cancer pain; they have also been advocated for CNMP. Opioids can have distressing side effects as dosages increase, such as constipation, sedation, drowsiness, nausea, decreased concentration and memory, or mood changes.[3] Most people who use opioids develop tolerance to the painkilling effect of opioids, and some become dependent on them. Studies have shown that high opioid usage can also put people's lives at risk.[4] Despite this, the prescription of opioid medication for CNMP has risen sharply in the higher income countries. Few studies of opioids have shown effectiveness beyond 12 weeks of follow-up. Population surveys have shown long-term use to be associated with increased side effects and limited pain relief.[3 5 6]

This synthesis of qualitative research was undertaken to underpin a process evaluation

## INTRODUCTION

Chronic non-malignant pain (CNMP) affects between an estimated 11% and 20% of the population in Europe and USA and

for the I-WOTCH (Improving the Wellbeing of people with Opioid Treated CHronic pain) study. I-WOTCH is a randomised controlled trial evaluating a multicomponent education and patient-centred group intervention with a one-to-one tapering programme against a control of an advice booklet with a relaxation CD. More information can be found in the main study protocol[7] and process evaluation protocol.[8] This qualitative evidence synthesis uses the methods of meta-ethnography to find out what people's experiences are of both using opioids for CNMP and their attempts to stop taking them.

## METHODS

We use Noblit and Hare's[9] seven stages of meta-ethnographic analysis. We used the new Meta-Ethnography Reporting Guidelines (eMERGe) to structure our report[10] (see online supplementary appendix 1). The protocol is published in the International Prospective Register of Systematic Reviews (PROSPERO) (http://www.crd.york.ac.uk/PROSPERO).CRD42017082418

### Step 1: getting started

In order to address what has been labelled an opioid epidemic,[11] we need to understand people's experiences of being on opioids and of coming off them. Our team was chosen because of its expertise in primary qualitative research and qualitative evidence synthesis specific to chronic pain and opioid prescription.

### Step 2: deciding what is relevant

We undertook systematic electronic searches in June 2017, with a rerun in September 2018, appraising relevant papers for quality using the Critical Appraisal Skills Programme (CASP) tool for qualitative research.[12] One researcher (VPN) with the assistance of an academic librarian (SJ) searched seven electronic databases: Medline, Embase, Allied and Complementary Medicine Database (AMED), Cumulative Index to Nursing and Allied Health Literature (CINAHL), PsycINFO, Web of Science and Scopus (Science Citation Index and Social Science Citation Index). Forward citation searches were also conducted. We used search terms, free text and medical subject headings (MeSH) terms for all opioid drugs as well as their generic names. We combined these with the MeSH term 'pain' and a wide range of MeSH terms and words to describe all types of qualitative research and its analysis based on a search used by Toye et al in 2017.[13] The search was limited to those in English and on humans, with no cut-off date. Online supplementary appendix 2 shows an example of our search terms.

Unique citations were screened independently by two researchers (VPN and ST; see the Acknowledgements section) against our inclusion and exclusion criteria (see box 1). Any disagreements were arbitrated by a third researcher (KS). Papers for full-text reading were identified and read by two researchers. Quality was assessed using the CASP tool. VPN critically appraised the studies

### Box 1   Inclusion and exclusion criteria

**Included studies.**
► Adults (18 years or older) taking or have taken opioid medication in the last 5 years.
► Published in English in peer-reviewed journals with no time constraints.
► Must relate to patient perspectives on using opioid medication for chronic non-malignant pain.
► Must use qualitative methodology (any analytical approach) or mixed quantitative and qualitative methodology with qualitative findings reported separately.
► Where studies include participants with differing medication, we will include studies where the experience of those taking opioids is reported separately.

**Excluded studies.**
► Paediatric studies (age less than 18 years).
► Theoretical or methodological papers.
► Purely quantitative studies or mixed methods studies where the qualitative data are not presented separately.
► Studies concerning active cancer.
► Studies concerning headache.
► Studies concerning any acute, or acute postoperative, pain.
► Studies concerned only with healthcare professional or carer perspectives, or studies of mixed carer/patient/professional populations where patient perspectives are not presented separately.
► Non-English-language studies.
► Theses or conference abstracts which are not peer-reviewed.

and KS independently appraised 10% for consistency. The CASP scores are shown in table 1 and online supplementary appendix 3. The GRADE-CERQual (Grading of Recommendations Assessment, Development and Evaluation working group - Confidence in Evidence from Reviews of Qualitative research) was used to appraise the reviewers' confidence in the research findings.[14 15]

### Step 3: reading the studies

VPN read all the studies, and KS and FT read half of these papers each (so all were read twice), and all extracted the second-order concepts independently. A second-order concept is a researcher's interpretation of data in a primary qualitative study.[16] VPN, KS and FT met to discuss and reach agreement, and compiled a spreadsheet of all of the concepts extracted from the papers.

### Step 4: determining how the studies are related

VPN sorted the concepts into categories by looking for any similarities and differences across all the studies. VPN, KS and FT discussed the categorisation of data on multiple occasions. To enable comparison across studies, VPN recorded descriptive data about each study (see table 1).

### Step 5: translating studies into each other

Patterns and associations between categories were explored, and all researchers felt that a line of argument approach as defined by Noblit and Hare[9] would be the most useful method to interpret the data.

**Table 1** Study characteristics with CASP and GRADE-CERQual relevance ratings

| Studies | Country | Data collection/participants | Analytical approach | Aims (text in italics indicates verbatim quotes) | Morphine equivalent daily dose mg/day (MED) | CASP score | Relevance |
|---|---|---|---|---|---|---|---|
| 1. Arnaert and Ciccotosto 2006[30] | Canada | Semistructured interview (n=11) (4M/7F) | Content analysis. | "…to develop an understanding and to gain knowledge about the beliefs of patients with CNMP who were taking methadone, and to explore what challenges, if any, existed that affected their beliefs when first starting methadone treatment as an activity in their personal lives." | None reported. | 17/20 | P |
| 2. Bergman et al 2013[29] | USA | Indepth interviews (n=26) (24M/2F) | Inductive thematic analysis. | "… to develop a better understanding of the respective experiences, perceptions, and challenges both patients with chronic pain and PCPs face communicating with each other about pain management in the primary care setting." | None reported. | 17/20 | P |
| 3. Blake et al 2007[17] | UK | One focus group (n=4) (2M/2F) and interviews (n=10) (3M/5F) | Interpretative phenomenological analysis. | "…to determine the attitudes and experiences of patients receiving long-term strong opioid medication for chronic non-cancer pain in primary care." | Individual opioid dosages. | 19/20 | R |
| 4. Brooks et al 2015[20] | Canada | Indepth interviews (n=9) (4M/5F) | Interpretative phenomenological analysis. | "…to explore the lived experience of adults using prescription opioids to manage CNCP, focusing on how opioid medication affected their daily lives." | None reported. | 17/20 | R |
| 5. Buchbinder et al 2015[39] | USA | Audio-recorded clinical encounters (n=74) (37M/37F) | Qualitative approach based on conversational analysis. | "We examined the direct and indirect means by which patients express a desire for analgesic medication." | None reported. | 18/20 | I |
| 6. Chang et al 2011[40] | USA | Face-to-face interview (n=21) (13M/8F) ≥65 years | Content analysis, mixed methods. | "… to: (1) describe older adults' patterns of adherence to their prescription opioid medication regimens and their reasons for these medication use patterns; and (2) examine the associations between adherence of prescription opioids, pain intensity, and pain interference on daily activity." | None reported. | 16/20 | P |
| 7. Chang and Ibrahim 2017[31] | Canada | Patient interviews (n=13) (1M/12F) | Not specified, no references given. | "… to explore the perceptions of patients with chronic pain and their physicians on OxyContin discontinuation and the impact that it had on chronic pain care and management." | None reported. | 16/20 | P |
| 8. Coyne et al 2015[41] | USA | Cognitive interview, semistructured, think-aloud approach (n=66) (27M/39F) | Content analysis. | "….to evaluate the content validity of the Stool Symptom Screener by assessing patient understanding of its items, patient ability to differentiate among response options, and patient perceptions about the use of a 2 week recall period for evaluating stool symptoms, BMs, and laxative use.<br>- to evaluate how patients describe their constipation experience and to understand whether this differs between patients who frequently use laxatives and those who do not." | None reported. | 17/20 | I |

**Table 1** Continued

| Studies | Country | Data collection/participants | Analytical approach | Aims (text in italics indicates verbatim quotes) | Morphine equivalent daily dose mg/day (MED) | CASP score | Relevance |
|---|---|---|---|---|---|---|---|
| 9. Esquibel and Borkan 2014[42] | USA | Indepth interviews with patients with CNCP receiving opioids and their physicians (n=21 patients) (8M/13F) | Immersion/crystallisation process to generate a thematic codebook. Mixed methods with some quantitative data analysis. | "…to better understand the effects of COT [chronic opioid therapy] on the doctor–patient relationship." | None reported. | 20/20 | R |
| 10. Frank et al 2016[23] | USA | Semistructured interviews (n=24) (11M/13F) | Team-based, mixed inductive and deductive approach guided by the health belief model. | " … to explore patients' perspectives on opioid tapering." | MED: used algorithm. Median (IQR) 70 (30–165), range 15–1845. | 20/20 | R |
| 11. Green et al 2017[43] | USA | 8 focus groups (only 2 with patients with chronic pain) (n=15) (8M/7F) | Thematic analysis. Content analysis with a collaborative codebook development process. | "…to explore the attitudes of various stakeholders toward pharmacy-based naloxone and opioid medication safety and to capture the range of initial experiences with pharmacy naloxone in 2 states with pharmacy naloxone policies." | None reported. | 18/20 | P |
| 12. Hooten et al 2011[44] | USA | n=18 Interviews (n=15) and focus group (n=3) (10M/8F) | Participatory research, mixed methods—thematic and content analysis. | "…to determine the attitudes and beliefs of both patients and pain medicine physicians regarding smoking cessation interventions applied during pain therapy." | MED: used equianalgesic conversion software program. Mean±SD 227±356. | 18/20 | I |
| 13. Krebs et al 2014[27] | USA | Semistructured interviews (n=26 patients) (13M/13F) | Immersion crystallisation approach. | "…to better understand primary care physicians' and patients' perspectives on recommended opioid management practices and to identify potential barriers and facilitators of guideline concordant opioid management in primary care." | None reported. | 16/20 | P |
| 14. Matthias et al 2014[45] | USA | Audio recordings of primary care visits and semistructured interviews after clinic visit (n=30) (26M/4F) | Immersion/crystallisation approach. | "…to advance understanding about communication about opioids by directly capturing clinical communication about opioids, as well as interviewing patients to gain insight into communication patterns and their broader relationships with their physicians, and how these relationships shape communication about opioids." | None reported. | 18/20 | R |
| 15. McCrorie et al 2015[25] | UK | Semistructured interviews with patients (n=23) (6M/17F) | Grounded approach for thematic analysis. Constant comparison. | " … to understand the processes which bring about and perpetuate long-term prescribing of opioids for chronic, non-cancer pain. We interviewed patients and GPs about their experiences, beliefs and expectations of analgesia prescribing, to improve understanding of how problematic long-term opioid prescribing becomes established". | None reported. | 17/20 | R |
| 16. Mueller et al 2017[22] | USA | Semistructured interview (n=24) (8M/16F) | Ethnographic iterative methods. Inductive and deductive approaches. | "This study assessed the knowledge and attitudes toward naloxone prescribing among non-cancer patients prescribed opioids in primary care." | Inclusion criteria ≥100 mg MED. | 16/20 | I |

Continued

**Table 1** Continued

| Studies | Country | Data collection/ participants | Analytical approach | Aims (text in italics indicates verbatim quotes) | Morphine equivalent daily dose mg/day (MED) | CASP score | Relevance |
|---|---|---|---|---|---|---|---|
| 17. Paterson et al 2016[21] | Australia | Qualitative individual interviews (n=20) | Constant comparison and decision making explored in depth and with a thematic analysis using a published 'Model of medicine-taking'. | "...to identify the varying influences on patients' decisions about their use of prescribed long-term opioids. The research questions are: 1) does this conceptual Model of medicine-taking apply to people taking prescribed opioids, 2) is the concept of "resistance" to medication useful in this context, and 3) does this concept lead to new insights which may improve communication and shared decision making between." | None reported. | 18/20 | P |
| 18. Penney et al 2017[28] | USA | 11 focus groups (n=80) and individual interviews (n=10) (27M/63F) | Theme identification | "...to identify the practical issues patients and providers face when accessing alternatives to opioids, and how multiple parties view these issues." | None reported. | 13/20 | P |
| 19. Rieb et al 2016[34] | Canada | Inperson, semistructured interviews (n=21) (14M/7F) | Deductive and inductive approaches. Use of survey and emergent interview themes. | "....to document the existence and characteristics of this pain phenomenon that we have named withdrawal-associated injury site pain (WISP)." | Recalled dose before WISP. | 18/20 | P |
| 20. Simmonds et al 2015[46] | USA | Semistructured focus groups ×3 (n=25) (17M/8F) | Grounded theory-informed approach. Framework provided by the theory of planned behaviour. | "...to examine barriers and facilitators to multimodality chronic pain care among veterans on high-dose opioid analgesics for chronic non-cancer pain." | Inclusion criteria at least 50mg MED. | 16/20 | P |
| 21. St Marie 2016[47] | USA | Semistructured interviews (n=12) (6M/6F) | Thematic and interpretive analyses. | "...to provide a deeper understanding of the experiences, issues, and challenges that people face who live with chronic pain and received opioids to help manage their pain in primary care. The following research questions were addressed: (a) What are the experiences of individuals who live with chronic pain and receive opioid pain medications to manage their pain in primary care? (b) What have been their healthcare experiences as they strive to manage their pain?" | None reported. | 19/20 | R |
| 22. Vallerand and Nowak 2009[19] | USA | Indepth serial interviews (n=22) (6M/16F) | Phenomenological study/ constant comparative method. | "...to examine the lived experience of adults receiving opioid therapy for relief of chronic non-malignant pain through the examination of data obtained through serial tape-recorded interviews." | Range 22.5-3200. | 14/20 | R |
| 23. Vallerand and Nowak 2010[24] | USA | Indepth serial narrative interviews (n=22) (6M/16F) | Phenomenological study/ constant comparative method. | "...to elucidate the essence of the lived experience of patients receiving chronic opioid therapy for chronic non-malignant pain through examining their life narratives." | None reported. | 13/20 | R |
| 24. Wallace et al 2014[48] | USA | Photovoice photos, interviews (first, n=31; second, n=25) and focus groups (n=19) | Grounded theory. | "...this study examined the utility of a combination of qualitative methods (Photovoice, one-on-one interviews, and focus groups) in examining the daily experiences of primary care patients living with chronic pain". | None reported. | 16/20 | R |

Continued

**Table 1** Continued

| Studies | Country | Data collection/participants | Analytical approach | Aims (text in italics indicates verbatim quotes) | Morphine equivalent daily dose mg/day (MED) | CASP score | Relevance |
|---|---|---|---|---|---|---|---|
| 25. Warms et al 2005[26] | USA | Comments written in the margins of a questionnaire (n=797) | Content analysis. | "…to determine the characteristics of those who wrote comments [written in the margin of survey questionnaire] and to understand what was being communicated in their comments." That comments were written in the margin is a potential limitation. | None reported. | 16/20 | P |
| 26. Zgierska et al 2016[49] | USA | Qualitative data on treatment satisfaction and experience (n=17) | Qualitative analysis methods. Grounded theory. | "… to determine feasibility, acceptability, and safety of an MM-based intervention in patients with CLBP requiring daily opioid therapy." | Inclusion criteria ≥30mg MED. Mean±SD 166.9±153.7. | 18/20 | I |
| 27. Zheng et al 2013[18] | Australia | Indepth qualitative interviews (n=20) (10M/10F) | Illness narratives. Thematic analysis. | "…to investigate the progression of the illness and opioid journeys of people who are taking opioids for chronic non-cancer pain". | None reported. | 17/20 | P |
| Rerun of search | | | | | | | |
| 28. Al Achkar et al 2017[32] | USA | n=9 (3M/6F) | Inductive emergent thematic analysis. | "…to evaluate the impact of Indiana's opioid prescription legislation on the patient experiences around pain management." | None reported. | 18/20 | R |
| 29. Matthias et al 2017[33] | USA | n=37 (12M/25F) | Inductive approach, constant comparison. | "…to understand communication processes related to opioid tapering." | None reported. | 18/20 | R |
| 30. Matthias et al 2018[50] | USA | n=34 (28M/6F) | Inductive approach, constant comparison. | "…to understand patients' experiences with the SPACE trial including their views and anticipated benefits of opioids, vs non-opioids, experiences with the intervention and to what extent expectations were met after completing the study." | None reported. | 19/20 | R |
| 31. Smith et al 2018[51] | USA | n=15 (4M/11F) | Concurrent, embedded, mixed methods design, Heideggerian hermeneutic phenomenological approach. | "This research presents an interpretation of the experience of seeking pain relief for a group of people taking opioid pain medications whose pain is not adequately controlled." | None reported. | 20/20 | P |

GRADE-CERQual relevance component: I, indirect relevance; P, partial relevance; R, relevant; U, uncertain relevance; BM, bowel movements; CASP, Critical Appraisal Skills Programme; CLBP, Chronic low back pain; CNCP, Chronic noncancer pain; CNMP, chronic non-malignant pain; F, female; GP, general practitioner; GRADE-CERQual, Grading of Recommendations Assessment, Development and Evaluation working group - Confidence in Evidence from Reviews of Qualitative research; M, male; MM, mindfulness meditation; PCP, primary care providers; SPACE, Strategies for Prescribing Analgesics Comparative Effectiveness.

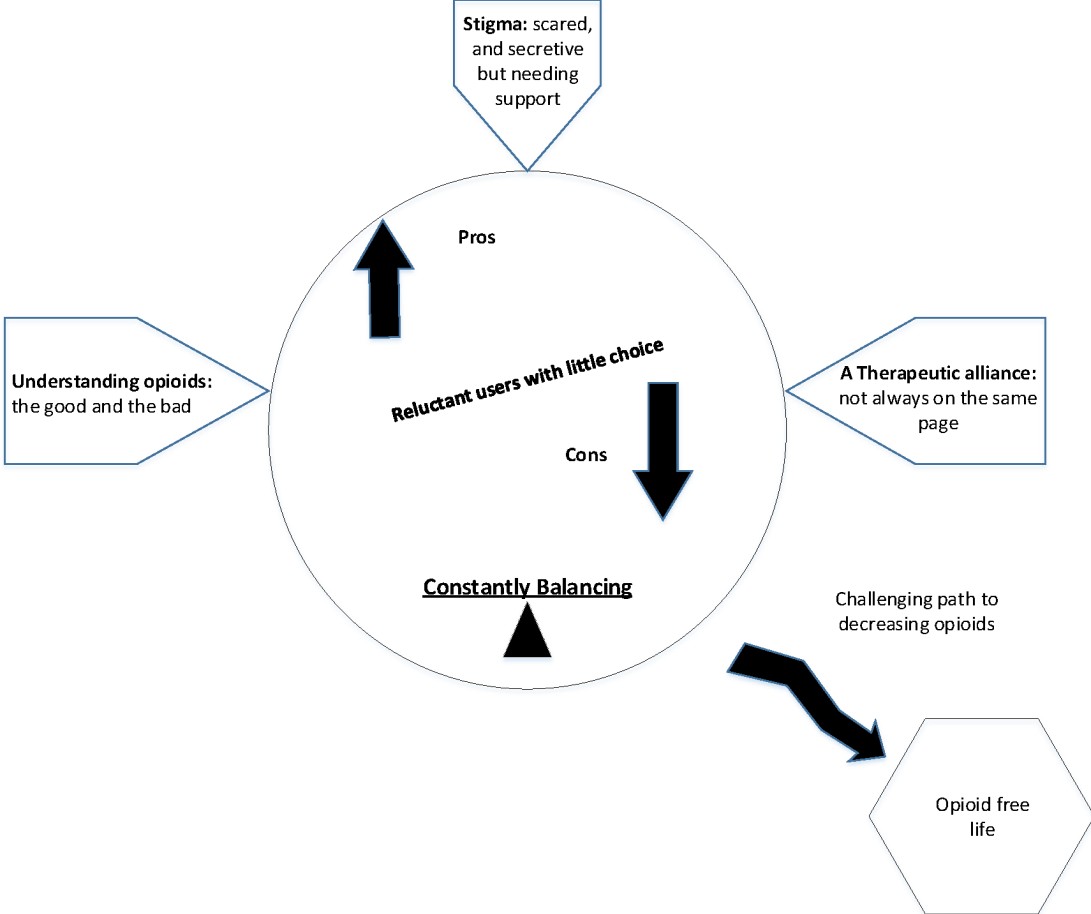

**Figure 1** Concept model of the experiences of people taking opioid medication for chronic non-malignant pain.

### Step 6: synthesising translations
Agreement was reached by clearly defining the overarching or third-order concepts arising from the data. A third-order concept is the reviewer's interpretation of second-order concepts.

### Step 7: expressing the synthesis
We developed a conceptual model to show how the themes related to each other in a line of argument (see figure 1).

### Patient and public involvement
We did not involve patients or the public in our work.

### RESULTS
Two reviewers (VPN and ST) screened 2994 titles or abstracts (after the removal of duplicates from the 5064 citations retrieved) and identified 153 full texts of interest. Two reviewers (VPN and KS) read these and 122 were excluded. Reasons are given in the Preferred Reporting Items for Systematic Reviews and Meta-Analyses flow chart (see figure 2). The reviewers agreed to include 31 studies. The included studies were from the USA (23), Canada (4) UK (2) and Australia (2), and used a range of qualitative methods.

We report the four facets of GRADE-CERQual for all papers: (1) methodological, (2) confidence, (3) relevance and (4) adequacy of data (see tables 2 and 3).

### Synthesis of findings
We abstracted five themes from the second-order concepts. Table 4 shows how each study contributed to each theme. We have illustrated each concept with exemplary quotations.

### Reluctant users with little choice
This describes a resistance or hesitancy to take opioids mainly due to concerns about side effects or addiction, although they felt there were no other options available.

> I don't want to become addicted, if I'm going to become addicted then as far as I'm concerned I'm a druggie, so I might as well not be here anyway, so I don't want to become addicted…. (Blake *et al*, p103)[17]

> I just didn't want to go on them because I mean once you get on them that's it, you're sort of stuck on them. I didn't want to take morphine at first because there was a girl that I went through one of the courses with and she always seemed really dopey and drugged up so it took them a long while to talk me to into taking

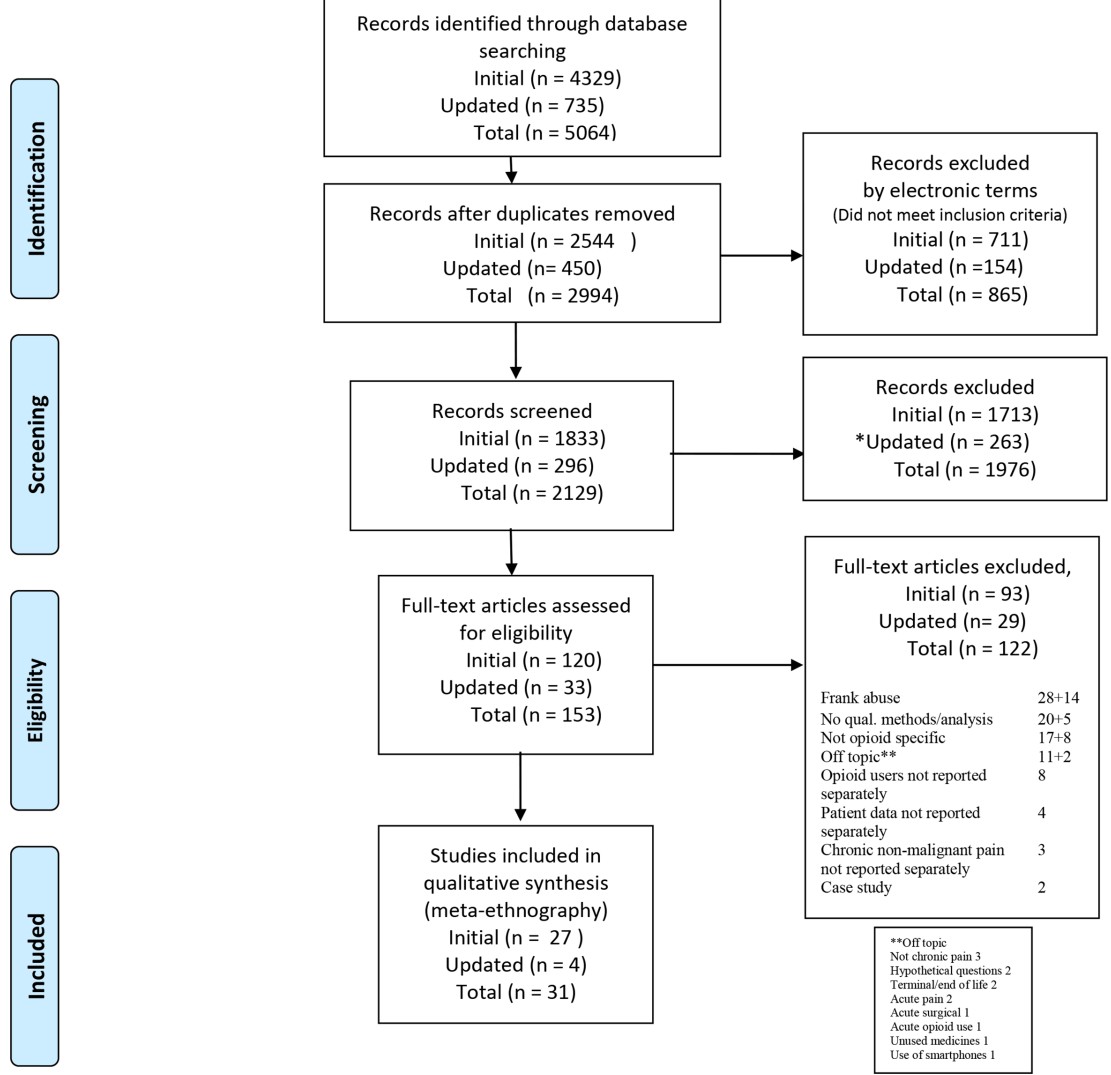

**Figure 2** PRISMA flow diagram.[52] PRISMA, Preferred Reporting Items for Systematic Reviews and Meta-Analyses.

the morphine because I didn't want to be like that. (Zheng et al, p1832)[18]

Some spoke of underusing or were keen to reduce their medications when possible. There was a dislike of being on long-term medication and some thought that it would not relieve their pain.

I don't want to do that [take more morphine]. I want to stay on as little as I possibly can because there might come a time when I need more and I don't want to be on high doses. I've always tried to keep it at a minimum amount of tablets each day…. (Blake et al, p105)[17]

Even though some were reluctant, there were other instances of dramatic improvement in people's lives. This then weighted their choice to stay on the opioids.

I mean it is just like a miracle as far as I am concerned. It is like knowing it [the pain] is there but you have the instruments to prevent it from getting

out and [be]coming a roaring demon. (Vallerand and Nowak, p170)[19]

But opiates, that's my way of life. There would be no life if I didn't have this. And I thank God for them because without them I'd be…well I wouldn't be. I just couldn't go on. I would have committed suicide a long time ago. And I say that truthfully cause you could not live like that, with that constant, constant pain. But, with the opiates it's made it possible to be able to have a part of a life, you know. (Brooks et al, p20)[20]

### Understanding opioids: the good and the bad
This describes patients' knowledge or understanding about opioids which had generally been acquired ad hoc and slowly over time, from pharmacists, patient package inserts in their medication, leaflets, the internet, television programmes and from doctors, especially doctors at the pain clinic.

**Table 2** Confidence in review findings: GRADE-CERQual assessment

| Review findings | Studies contributing (see table 1 column 1 for study number) | Methodological limitations (study number) | Relevance (see table 1 end column) | Coherence | Adequacy of data |
|---|---|---|---|---|---|
| Reluctant users with little choice | 1, 3, 4, 5, 6, 7, 17, 18, 21, 22, 26, 27, 30 (13 studies) | 11 no concerns. 2 minor concerns (18, 22). | 5 Relevant. 6 Partial. 2 Indirect. | No concerns. | No concerns. |
| Understanding opioids: the good and the bad | 1, 3, 7, 9, 10, 11, 15, 16, 17, 23, 25, 27, 29 (13 studies) | 12 no concerns. 1 minor concerns (23). | 6 Relevant. 6 Partial. 1 Indirect. | No concerns. | No concerns. |
| A therapeutic alliance: not always on the same page | 1, 2, 3, 4, 5, 7, 9, 10, 11, 13, 14, 15, 16, 17, 18, 19, 20, 21, 22, 23, 24, 25, 26, 28, 29, 31 (26 studies) | 23 no concerns. 3 minor concerns (18, 22, 23). | 12 Relevant. 11 Partial. 3 Indirect. | No concerns. | No concerns. |
| Stigma: feeling scared and secretive but needing support | 1, 2, 3, 4, 7, 9, 10, 14, 16, 17, 18, 20, 21, 22, 23, 24, 27, 28, 31 (19 studies) | 16 no concerns. 3 minor concerns (18, 22, 23). | 10 Relevant. 8 Partial. 1 Indirect. | No concerns. | No concerns. |
| The challenge of tapering/withdrawal from opioids | 7, 10, 18, 19, 30, 31 (6 studies) | 5 no concerns. 1 minor concerns (18). | 2 Relevant. 4 Partial. | Minor concerns. | Minor concerns. |

GRADE-CERQual, Grading of Recommendations Assessment, Development and Evaluation working group - Confidence in Evidence from Reviews of Qualitative research.

When you see it in the media, when you see it on the television, you think if you're taking regular morphine you must be in a pretty bad way, you know. (Blake *et al*, p103)[17]

I always ask before I go on a medication, what are the side effects, I was told I may experience constipation; nothing else was explained to me. (Paterson *et al*, p721)[21]

There was often poor knowledge about using opioids for chronic pain, and about addiction, overdose risk and side effects.

There's not too much education about it [overdose]…When I first started taking it [the opioid medication], no one told me about OD [overdose] or anything about that. Because I was taking it not [as] prescribed…I was just like when I felt pain I would just take like five or six of them or whatever. Then at the end, I'd run out. (Mueller *et al*, p279)[22]

Patients often had to defend their usage and this added to their stress especially when they felt their healthcare professionals lacked an understanding of the place for

**Table 3** GRADE-CERQual component scoring

| Methodological limitations. Coherence. Adequacy of data. | No or very minor concerns. Minor concerns. Moderate concerns. Serious concerns. |
|---|---|
| Relevance. | Relevant. Partial. Indirect. Uncertain. |

GRADE-CERQual, Grading of Recommendations Assessment, Development and Evaluation working group - Confidence in Evidence from Reviews of Qualitative research.

opioids in the treatment of CNMP or were cautious about using them.

The concern is that if they increase my opioid dosage, I could stop breathing. It's ridiculous. (Frank *et al*, p1841)[23]

There are still a lot of doctors out there that are against it. They think it is bad. Bad medicine. Bad practice. (Vallerand and Nowak, p129)[24]

In contrast, some people felt well informed, which either produced more concern or gave the patients confidence in their opioid regimen.

… and from what I've read up, because I like to, sort of, keep on top of things, that it's an opium based drug, so you will build up some tolerance and you will build up [becomes tearful] And you will potentially become sort of addicted to it, if you like. (McCrorie *et al*, p3)[25]

Under Dr A [pain clinic] I've learnt more. And my concern has been, well it was initially the possibility of addiction, but she has assured me that I'm not showing no signs of addiction at all. I may have some withdrawal problems. (Paterson *et al*, p721)[21]

### A therapeutic alliance: not always on the same page

This describes a therapeutic alliance or the relationship between patients and their healthcare providers, which was considered important.

Overall there was a feeling that healthcare professionals and patients were often 'not on the same page' about opioid usage.

My family doctor…does not want me to be dependent on heavy pain meds, so I am intensely miserable 99% of the time. (Warms *et al*, p252)[26]

**Table 4** Themes apparent in each study

| Author | RU | U | TA | S | TW |
|---|---|---|---|---|---|
| 1. Arnaert and Ciccotosto[30] | X | X | X | X | |
| 2. Bergman et al[29] | | | X | X | |
| 3. Blake et al[17] | X | X | X | X | |
| 4. Brooks et al[20] | X | | X | X | |
| 5. Buchbinder et al[39] | X | | X | | |
| 6. Chang et al[40] | X | | | | |
| 7. Chang and Ibrahim[31] | X | X | X | X | X |
| 8. Coyne et al[41] | | | | | |
| 9. Esquibel and Borkan[42] | | X | X | X | |
| 10. Frank et al[23] | | X | X | X | X |
| 11. Green et al[43] | | X | X | | |
| 12. Hooten et al[44] | | | | | |
| 13. Krebs et al[27] | | | X | | |
| 14. Matthias et al[45] | | | X | X | |
| 15. McCrorie et al[25] | | X | X | | |
| 16. Mueller et al[22] | | X | X | X | |
| 17. Paterson et al[21] | X | X | X | X | |
| 18. Penney et al[28] | X | | X | X | X |
| 19. Rieb et al[34] | | | X | | X |
| 20. Simmonds et al[46] | | | X | X | |
| 21. St Marie[47] | X | | X | X | |
| 22. Vallerand and Nowak[19] | X | | X | X | |
| 23. Vallerand and Nowak[24] | | X | X | X | |
| 24. Wallace et al[48] | | | X | X | |
| 25. Warms et al[26] | | X | X | | |
| 26. Zgierska et al[49] | X | | X | | |
| 27. Zheng et al[18] | X | X | | X | |
| **Studies from search rerun** | **RU** | **U** | **TA** | **S** | **TW** |
| 28. Al Achkar et al[32] | | | X | X | X |
| 29. Matthias et al[33] | | X | X | | X |
| 30. Matthias et al[50] | X | | | | |
| 31. Smith et al[51] | | | X | X | |

RU, reluctant users with little choice; S, stigma: feeling scared and secretive but needing support; TA, a therapeutic alliance: not always on the same page; TW, the challenges of tapering or withdrawal; U, understanding opioids: the good and the bad; X, theme present in the paper.

Some patients felt they were not listened to and were frustrated by a lack of empathy from physicians regarding their pain experience.

I frequently have difficulty with the residents (doctors in training) explaining why these drugs, this many drugs…Finally Dr. [family physician] wrote a note in my file – stop harassing [participant's name]. This is what she gets and why she gets it. And they did stop but it was inconvenient. For instance, they would not prescribe me three months at a time. I would be dispensed one month at a time. And for someone who had been taking the same drugs for 10 years I found that condescending. (Brooks et al, p18)[20]

A reluctance to prescribe from general practitioners and pharmacists and the use of opioid contracts or a restriction of medication were often considered punitive.

It kind of made me feel like I was doing something wrong, which I wasn't, but I signed a contract. You know, what would I be without my meds? (Krebs et al, p1152)[27]

And I told my doctor that, that I wanted so I could sleep through the night. And now he, well, I'll give you 10, but it's got to last. Like he treats me like a drug addict. (Penney et al, p6)[28]

The healthcare system often worked against a therapeutic alliance with lack of continuity or care or frequent

visits, which fed into mistrust. Patients complained that provider turnover affected their ability to receive individualised care; conversations about pain and treatment options often had to be started over again from scratch.

I don't have the same doctor long enough to know. (Bergman *et al*, p1693)[29]

However having blood or urine tests for levels of opioids and regular checks were seen by some as being cared for.

I would say, 'I have this agreement and you don't have to sign it if you don't want to, but I would like to go over it with you. These are suggestions because this medication is addictive, it is dangerous, and I just want to make sure you're aware.' I think if you really want to make it where people are not hostile, say they have to have a urine test every 6 months, everybody, and that 'it's a policy because we care about all of you.' (Krebs *et al*, p1152)[27]

Some talked of the need for good relationships built on trust, shared decision making and knowledgeable specialists who communicate well.

I wouldn't say I researched it to that depth, you know, I read a little bit about, and asked a lot of questions at my doctor, and then we decided. (Paterson *et al*, p722)[21]

### Stigma: feeling scared and secretive but needing support

This describes feelings of stigma and fear which people expressed directly in relation to their opioid usage. This includes people's negative attitudes from family, medical professionals and work colleagues which lead to them feeling stigmatised and judged for taking opioids.

So I'm constantly trying to clean up because I think people are going to judge me. 'Oh, because she's on all this medication, ooh, she can't look after her children.' (Paterson *et al*, p724)[21]

As soon as you mention to someone that you are on pain medication it's, 'Oh my god, you've got to get off it.' It is viewed as weak. Somehow I am weak for being on this medication. (Vallerand and Nowak, p128)[24]

To protect themselves, some chose to keep their opioids a secret.

But you know, after 2 years of pain, you are physically exhausted, mentally exhausted and depressed. So, I take my medication and I hide it at the bottom of my drawer. It's my secret life. It's always a secret, and I've got to hide it and not tell anyone. (Vallerand and Nowak, p169)[19]

Some people made a conscious decision about who they could tell and who they could not due to negative reactions. Relationships suffered when patients felt unsupported.

My son told me I was a drug addict. He did. He really did. He was to the point, he didn't know what he could do for me. It really was that bad. (Vallerand and Nowak, p128)[24]

I had originally told my sister and she was very concerned. Then she said, 'As long as you don't stay on them.' She thought it was OK if I did it for a while but as long as I didn't stay on them. So I just sort of never told her. And she never asked. (Vallerand and Nowak, p128)[24]

Although some seemed confident in using opioids, mostly people spoke about fears such as addiction and uncontrolled pain. Feeling supported validated their choices and experiences and lessened some of their fears and concerns.

And at the end, my partner says—we sat down there and he goes 'Stay on them.' …I've always spoke to my partner, and if he's been unsure—we've both been unsure, we've both gone into the doctor together to ask questions. (Zheng *et al*, p1834)[18]

my wife wanted me to take this medication. She was like: let's go for it. (Arnaert and Ciccotosto, p26)[30]

### The challenge of tapering/withdrawal from opioids

Four papers[23] [31–33] explored patients' experiences of tapering or withdrawing as their main content. Two further papers[28] [34] addressed it as a more peripheral issue (see CERQual ratings in table 2). This describes the challenges and profound effects of tapering or withdrawing from opioids.

Tapering and withdrawing from opioids could be challenging and provoke anxiety.

I have a tremendous fear in a doctor saying I want you to taper off the methadone and get totally off the methadone with no alternative whatsoever. I think that would be an irrational decision by a doctor, and I probably wouldn't take that advice. (Frank *et al*, p1842)[23]

This anxiety could be alleviated by support from a trusted healthcare provider or other person.

The best thing about it was that nobody acted like I was a bad person because I was on these medications and was having to be going through this really slow process of coming down off of them. (Frank *et al*, p1843)[23]

Successful tapering was described as a collaborative agreement between the healthcare professional and the patient.

She put me down to 2 and a half [pills per day]. Then she said, okay, we'll go down to half a pill. I told her I didn't think that just 2 a day would do it, and she said okay, we'll try 2 and a half, are you agreeable with that? I said that's fine. I mean, we can discuss stuff. It doesn't have to be a disagreement because we can

talk about it. It's not an argument. We're 2 adults having a conversation, figuring out what to do. (Matthias *et al*, p1368)[33]

However, not all people experienced joint decision making when tapering.

I just don't feel that he's understanding. he don't seem to care what I'm saying, because he's lowering it down anyway, even though I've told him…that I didn't agree with it being lowered. (Matthias *et al*, p1369)[33]

For those in the USA, prescribing policies, advising clinicians to monitor and decrease opioid use, and the legislation to enforce these policies made those taking opioids feel as if they were 'a public health problem'. This could have a negative effect on the doctor–patient relationship and leave the patient feeling disempowered. This was compounded when opioids had been withdrawn by legislation.[31 32]

I have to struggle, suffer, to make the next the next time that I can get my medicine. And I don't think that's fair to me because if I can take my medicine a little more regularly, I would be able to do more…I don't think that the law, people, politicians, or anybody should be able to tell anybody that's in pain what type of medicine they can take. (Al Achkar *et al*, p7)[32]

That kinda got me mad, cause I thought well you know…they're taking it off the market because of people abusing it…It's not fair to us, you know…I think the government was wrong to…pull them off the market, you know, because of people abusing them, no like they weren't looking at the people that need them…But I think it's really unfair that people that really do need them can't get them. (Chang and Ibrahim, 2017, p3)[31]

### Overarching theme: constantly balancing

After considering the fives themes, an overarching theme emerged: 'Constant balancing'. The theme *reluctant users with little choice* describes the need to balance the pros and cons of starting opioids and the need to balance having pain with their hesitancy to use opioids.

I don't really like being on a lot of tablets, I've never been a tablet person, um…but I mean I can't have the pain either so it's one evil outdoing the other evil. (Paterson *et al*, p723)[21]

Studies describe balancing the dose for pain management with their side effects to allow them to function. Participants constantly weighed up the effects on their life: dealing with an internal conflict of unresolved pain versus necessary medication, being opioid free versus having uncontrolled pain, and balancing other stressors against opioid dose changes.

If you're going to be able to walk, and you take one pain pill so you can walk and live life, you're going to do it, even though you may not like it. (Penney *et al*, p6)[28]

The theme *stigma: feeling scared and secretive but needing support* describes the need to balance their hopes for relief with fear of side effects, and also to balance whether or not to disclose their opioid use with the risk of being labelled a 'drug seeker' versus having unrelieved pain.

I do it for my own protection by not telling them because I see how they react by reading something in the paper…and it's just their ignorance. And I don't have time. Well they know what's going on but they don't get it to this day. So you have to pick your battles…. (Brooks *et al*, p19)[20]

The theme *understanding opioids: the good and the bad* showed people had different levels of understanding, but weighed up their decisions and trade-offs against their pain relief.

It's, it's got a good and bad side, morphine….When I take it, it works really, really well but it makes you feel rather sick, umm, rather spaced out and thinking wise, umm, it outcomes more on the other, do I want to be sick or do I want to cry with pain? So I'd rather be sick but it is a very, very good painkiller. (Blake *et al*, p105)[17]

The therapeutic alliance theme showed that often it was evident that they were 'not on the same page', with them balancing the advice from their doctors with what they wanted.

[My provider] said you could die any time, and my husband and I said, well, we realize that, but because of the pain, you know, we were willing to take that risk that I would die from the narcotic medication. (Frank *et al*, p1841)[23]

It also meant that there were multiple barriers to the process of decreasing opioids due to this constant balancing act, which is described in the theme *the challenge of tapering/withdrawal from opioids*.

I will tell her, if I do come off this medication, there are going to be consequences. I can't walk as often, I can't stand as long, I just can't do it…. (Vallerand and Nowak, p169)[19]

### DISCUSSION

Our five themes were (1) reluctant users with little choice; (2) understanding opioids: the good and the bad; (3) a therapeutic alliance: not always on the same page; (4) stigma: feeling scared and secretive but needing support; and (5) the challenge of tapering or withdrawal. An overarching theme of 'constantly balancing' emerged from the data. These themes all had positive and negative aspects, although the negative were more prevalent .

We present a line of argument of how complex it is for the patient to balance decisions at every stage of their journey. First is their reluctance to start taking opioids but feeling they had no option. Patients are given opioids for CNMP often as a last resort when all other treatment has failed and their lives are so profoundly affected that they talk of a desperation, that they would literally 'try anything'. Patients spoke about not being given any detailed information about opioids and that they had learnt more about them over time from different sources. This varied understanding about opioids and their side effects can affect the decisions that people make. Patients reported the need to keep the dosage of opioids as low as possible and often that they were not at risk of addiction or overdose if they were taking them as prescribed. Even those who felt they may be addicted sometimes viewed this as an acceptable trade-off for pain relief. Our findings indicate that patient desperation combined with inadequate information from healthcare professionals could trigger the prescription of opioids. It may be that delivering accurate information about the potential side effects and limited efficacy of opioids for chronic pain management would reduce the use of opioids.

Our findings demonstrate that the stigma surrounding how patients feel about being on opioids can be compounded by the judgements of others. Although patients often describe themselves in terms of 'reluctant users', if they experienced the benefits of opioids through decreased pain and thus increased function they are often too scared to reduce opioids and return to a life of potentially unmanaged pain.

Our findings suggest that clinicians and patients with chronic pain are not always 'on the same page'. The theme *a therapeutic alliance* captures the positives, but also the tensions and mismatches of perceptions held by healthcare providers who are attempting to limit dose escalation, and patients who may view constant dose escalation as an acceptable trade-off for reducing relentless pain. The therapeutic alliance is a robust theme supported by 26 of the 31 studies included. This is not surprising as patients rely on their healthcare professionals to prescribe opioids. This finding resonates with qualitative evidence syntheses exploring the experience of patients[13] and healthcare professionals.[35] It seems clear that joint decision making is important for appropriate healthcare; however, our findings suggest that there are instances of mistrust on both sides. A qualitative evidence synthesis exploring clinicians' experience of prescribing opioids for chronic pain demonstrates that the process of prescribing opioids is not straightforward for clinicians who face a complex decision: "'Should I shouldn't I' prescribe opioids for CNMP?"[35] They also demonstrate that clinicians must walk a fine line to balance the pros and cons of opioids while also maintaining patient trust. This suggests that both patients and healthcare professionals find dealing with prescribing/taking opioids for CNMP is complex and involves balancing and trade-offs.

Current guidance from the Royal College of Anaesthetists in the UK and the Centers for Disease Control and Prevention in the USA advocates a preference for non-opioid therapies in the treatment of CNMP.[36] If a clinician feels that opioids are indicated, then they recommend a low dose for a short duration, which should be assessed for effectiveness and regularly evaluated for benefits and harms. All but four studies in this review are between 2005 and 2017, prior to these guidelines. Opioid contracts in some areas of the USA and Canada can make patients feel stigmatised and judged, and this effect can be moderated by a good therapeutic relationship, and reframing these as agreements rather than contracts.[37] Some physicians may view contracts/agreements as necessary to guard against uncontrolled dose escalation, repeated demands for replacement of lost or misplaced medication, subversion and illicit opioid intake. This finding resonates with Toye *et al*[35], who described the moral boundary work and social guardianship that clinicians associate with opioid prescription. Our findings suggest that this role may not contribute to an effective therapeutic partnership.

### Limitations of this study

A majority of the studies are from the USA, and the findings need to be taken in the context of its health and social care systems. Most of the articles in this qualitative synthesis were published or the research was conducted before the impact of the opioid epidemic became clear to regulators and the medical profession. Some papers discussed using opioids as a last resort, although the opioid epidemic, especially in the USA, indicates that the threshold for prescribing opioids was low until recent initiatives to discourage prescribing long-term opioids for chronic pain.[38] Not all studies gave morphine equivalent data, so we cannot determine what proportion were taking high, medium or low doses. We acknowledge that our interpretation of the data might have been influenced by the current, much more critical perception of opioid use for CNMP. Further evidence is needed to find out if these themes are universal for developed countries or whether there are important differences.

Our conceptual framework highlights that patients need to constantly balance and to consider the pros and cons of taking opioids. This can have a profound effect on people's relationships with their family, friends and healthcare providers and their perceived standing in the community, which is reflected in their careful balancing of disclosure. The therapeutic alliance and having a clear understanding of all the positive and negative aspects of opioids were important factors that underpinned their ability to maintain this fragile balance. This balance might also affect a person's desire or ability to taper or withdraw from opioids.

The GRADE-CERQual ratings (table 2) revealed we had confidence in the findings, with only a few minor concerns and no moderate or serious concerns.

## CONCLUSIONS AND RECOMMENDATIONS FOR FUTURE RESEARCH

The first meta-ethnography on this topic revealed a constant balancing and a life in flux in an effort to maintain participation in life and relationships. These are important features of opioid use for CNMP. To maintain this delicate balance, they often need support from family or clinicians; however, this balance can be upset by the feeling of being judged by this same potential support system or peers and society at large through the media. The therapeutic alliance with healthcare professionals, the extent of people's understanding, as well as the stigma attached to opioid use need to be navigated by people who are often reluctant to be on opioids in the first place.

**Acknowledgements** We would like to thank Samantha Johnson, an academic librarian who helped with the electronic searches, and Dr Stephanie Tierney, who helped to screen the citations.

**Contributors** VPN and KS contributed to the review concept and design as part of the I-WOTCH process evaluation team. HKS, SE, MU and KS were involved in the design of the I-WOTCH study. VPN, KS and FT screened the search results or extracted the data, and conducted the analysis and synthesis. All authors contributed to data interpretation, revised the final manuscript critically for important intellectual content and appraised the final manuscript. VPN prepared the final manuscript and is the corresponding author.

**Funding** This project was funded by the National Institute for Health Research, Health Technology Assessment (project number 14/224/04). The views and opinions expressed therein are those of the authors and do not necessarily reflect those of the HTA, NIHR, NHS or the Department of Health.

**Competing interests** KS has undertaken other meta-ethnographies and was on the NIHR HS&DR Board until January 2018. SE is an investigator on a number of NIHR and industry-sponsored studies. He received travel expenses for speaking at conferences from professional organisations. SE consults for Medtronic, Abbott, Boston Scientific and Mainstay Medical, none in relation to opioids. SE is Chair of the BPS Science and Research Committee. SE is Deputy Chair of the NIHR CRN Anaesthesia Pain and Perioperative Medicine National Specialty Group. SE's department has received fellowship funding from Medtronic, as well as nurse funding from Abbott. HKS is Director of Health Psychology Services, providing psychological services for a range of health-related conditions. MU was Chair of the NICE accreditation advisory committee until March 2017, for which he received a fee. He is chief investigator or coinvestigator on multiple previous and current research grants from the UK National Institute for Health Research and Arthritis Research UK, and is a coinvestigator on grants funded by the Australian NHMRC. He is an NIHR senior investigator. He has received travel expenses for speaking at conferences from professional organisations hosting conferences. He is a director and shareholder of Clinvivo, which provides electronic data collection for health services research. He is part of an academic partnership with Serco related to return to work initiatives. He is a coinvestigator on a study receiving support in kind from Stryker. He has accepted honoraria for teaching from CARTA. He is an editor of the NIHR journal series, and a member of the NIHR Journal Editors Group, for which he receives a fee.

**Patient consent for publication** Not required.

**Ethics approval** Not applicable - a review

**Provenance and peer review** Not commissioned; externally peer reviewed.

**Data availability statement** Data are available on reasonable request.

**ORCID iD**
Vivien P Nichols http://orcid.org/0000-0002-3372-1395

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
