## [Reviewer comments · BMJ Open]

ARTICLE DETAILS

TITLE (PROVISIONAL)	A systematic review and qualitative evidence synthesis of the experiences of people taking opioid medication for chronic non-malignant pain: a meta-ethnography
AUTHORS	Nichols, Vivien; Toye, Francine; Eldabe, Sam; Sandhu, Harbinder; Underwood, Martin; Seers, Kate

VERSION 1 – REVIEW

REVIEWER	Barbara J Turner University of Texas Health Science Center San Antonio
REVIEW RETURNED	13-Aug-2019

GENERAL COMMENTS	BMJ open review In preparation for conducting another project, the authors performed a systematic assessment of qualitative evidence about patients' views regarding taking opioids. The search strategy appears to be comprehensive. The grading of the quality of the studies was also strength. The main themes identified from the final group of 31 studies seem to make sense but with significant caveats. The investigators claim to use meta-ethnography but, as described in this paper, it appears to be a relatively superficial analytic approach. The first major concern relates to heterogeneity of included studies. They include diverse types of patients such as subjects in clinical trials of nonpharmacologic approaches to managing chronic pain (Zgierska) and patients giving opinions about having to change their opioid dose (Al Achkar). Another study includes patients who have known substance use disorders who clearly have greater problems with opioids. The Warms et al study is not appropriate because it examines comments written in the margins of a questionnaire by patients who have had a spinal cord injury and amputation. In addition, two studies use the same cohort which does not make sense. This hodgepodge of studies significantly limits the value of this analysis. At a minimum, table 1 more specific about the opioid experience of included patients and, when available, the dose in morphine equivalent units. A second major concern reflects these authors' lack of access to the full set of the qualitative data from these studies. They cannot take the original data to arrive at overarching themes similar to a meta-analysis of quantitative data from multiple studies. Thus, it is unclear how the themes from these studies lead to the five themes that were identified. The figure that tries show how these themes relate to each other is simplistic. The paper would benefit from a table summarizing key themes identified by the authors of each of the relevant studies. Thus the reader could example more information from these studies. For example, several studies including Frank et al and Simmonds et al focus on the lack of
---

	understanding of nonpharmacologic approaches to managing chronic pain. This is an important, clinically relevant focus that is lost in the theme related to understanding opioids. Another theme that really isn't elicited relates to patient challenges with accessing opioids - this goes beyond tapering and withdrawal but reflects difficulties patients face with receiving refills of opioid medications on time due to a cumbersome healthcare system, poor communication, and multiple physicians. This was a clear message of several of these studies including Simmonds. The discussion has relatively few references and mostly reflects opinion. Some statements are not based in fact at least in regard to the US experience. For example, on page 29 lines 31 through 33 the authors state that patients are being given opioids as a last resort. This has not been the case in the United States and it is a major reason for the opioid epidemic. Another sentence that doesn't make sense is page 29 lines 51 through 52 on where it states that individuals who were concerned about addiction thought it was still reasonable to receive opioids for short-term pain relief. Most patients with chronic pain are looking to use these drugs long term and not for short term pain relief. Another concern is on page 31 lines 20 through 24 where the term opioid contracts is used. This is unacceptable language because these are not legal contracts and many papers have been written about their being agreements at best. If opioid agreements are addressed, the authors should reference ample literature regarding benefits and challenges. Minor comments CASP needs to be spelled out in the abstract Table 1 offers a direct quote of the aims of the included studies which seems superfluous. The comment on page 4 about the expertise of the team is not needed
--	---

REVIEWER	Shabbar I. Ranapurwala University of North Carolina at Chapel Hill, Department of Epidemiology
REVIEW RETURNED	04-Oct-2019

GENERAL COMMENTS	This is a very well written paper documenting the experiences of patients receiving opioid pain medications for chronic non-cancer pain patients. The authors did a great job synthesizing the literature into emergent themes that describe the challenges faced and support available for these patients. I have a few concerns, however, I think the authors may be able to clarify them.  1. It seems that there were about 41% duplicates - is this really true? Why was this? Was this because different systems were pulling the same papers? 2. It is unclear which different criteria were used to first exclude 865 records during identification process and then another 1976 during the screening process, please clarify in the manuscript and the PRISMA diagram. 3. In the PRISMA flow diagram, in the last box of included - please say "meta-ethnography" instead of meta-analysis. 4. In the eligibility exclusion (n=122), the 3 "not chronic pain" could also go under the "off topic." 5. Table 2 row 1 column 3, Add the six studies with concerns here and then say the remaining 7 have no concerns. 6. Table 2 row 1 column 4, what about the 13th study? Same goes for row 2, columns 3 and 4. 7. Table 2 row 3 column 4, what about the other 3 studies?
--

	8. Table 2 row 4 column 3, what about the other 5 studies? 9. Table 2 row 4 column 4, what about the other 2 studies? 10. Table 2 row 5 column 2, say "(6 studies)" here 11. Instead of using author initials in the methods, use person neutral language in the main manuscript. All author contributions can be listed in the acknowledgments. E.g., Step 3 could be rewritten as, "Three researchers read the studies such that each study was read at least once and all researchers extracted the second order concepts independently. All three researchers met to discuss and reach"
--	--

REVIEWER	Frank Petzke Universitätsmedizin Gottingen Zentrum Anaesthesiologie- Rettungs- und Intensivmedizin
REVIEW RETURNED	07-Oct-2019

GENERAL COMMENTS	The authors present a meta-ethnography summarizing and integrating qualitative studies about the experience of patients on opioid medication and/or coming off them. This current and timely analysis addresses the important perspective of chronic pain patients in the current discussion on the benefit and risks of chronic opioid treatment. The central theme of constantly balancing was concluded from the studies. The reviewer is not an expert on conducting meta-ethnography, but the methodology seems sound, thorough and exhaustive, and is described in detail (Noblit and Hare's 7 stages, grade-CERQual assessment, eMERGe meta-ethnography reporting) and presented accordingly. The study is well-written with clear conclusions. There are no major comments, but the authors may consider the following thoughts. The five themes evolving out of this analysis make sense from a clinical perspective and reflect the general ambivalence of both patients and prescribers. The phrasing, however, of the theme "Understanding Opioids: the good and the bad" seems not ideal, since the various and complicated processes of gaining knowledge is the focus and not the realisation of potential benefits, like functional or other improvements. It is interesting, that what now is the focus of most current guidelines, namely to ensure improvement in physical and social functions with opioid appears not as a relevant "good" aspect, except in the context of "the reluctant user". The authors include 31 studies with the vast majority coming from the US prior to current guidelines and the realisation of the impact of the opioid crisis. They rightly comment on this as a relevant limitation. However, as this period is commonly characterized as a period of uncritical prescribing of opioids, their findings seem somewhat counterintuitive by already highlighting ambivalence and risk. May be there also is an influence (as an additional limitation) of the current justifiably critical perception of opioid treatment on the process of developing the overarching concepts? Although the theoretical construct behind this meta-ethnography relies on qualitative analysis of content, background, treatment setting and other patient related factors play a relevant role in shaping this content. A lot of this information is already summarized in table 1. May be some additional information on the patients could be added as a separate column, similar to comments like methadone specific or USA Veteran. Or to highlight if none of this information is available.
--

VERSION 1 – AUTHOR RESPONSE

Reviewer: 1 R1-1 In preparation for conducting another project, the authors performed a systematic assessment of qualitative evidence about patients' views regarding taking opioids. The search strategy appears to be comprehensive. The grading of the quality of the studies was also strength.	Thank you for your comments. We are pleased that you highlighted these strengths
R1-2 The main themes identified from the final group of 31 studies seem to make sense but with significant caveats. The investigators claim to use meta-ethnography but, as described in this paper, it appears to be a relatively superficial analytic approach. The first major concern relates to heterogeneity of included studies. They include diverse types of patients such as subjects in clinical trials of nonpharmacologic approaches to managing chronic pain (Zgierska) and patients giving opinions about having to change their opioid dose (Al Achkar). Another study includes patients who have known substance use disorders who clearly have greater problems with opioids. The Warms et al study is not appropriate because it examines comments written in the margins of a questionnaire by patients who have had a spinal cord injury and amputation. In addition, two studies use the same cohort which does not make sense. This hodgepodge of studies significantly limits the value of this analysis.	These studies all give qualitative information about how people felt about being on opioids in a variety of different settings. Each paper met the eligibility criteria. The diversity of the settings and the aims of the studies has been taken into account with the GRADE -CERQual ratings specifically the Relevance category in table 1. The reason we included both studies with the same cohort was because they were analysed from a different perspective. The inclusion of all these studies adds to the richness of the data and is a strength because the very heterogeneity of the studies increases the possibility of transferability of the findings across multiple settings.
R1-3 At a minimum, table 1 more specific about the opioid experience of included patients and, when available, the dose in morphine equivalent units.	Thank you we agree. Of the included papers, 8 gave morphine equivalents; 4 reported them as patient characteristics with either median/IQR or Mean/ SD or the range. 3 gave ME's as eligibility criteria ≥ 30, 50 or 100mgs and one gave individual morphine dosages. Reporting was inconsistent and it is unclear how morphine equivalents were estimated. We have added a column to table one to give these where they were available.
R1-4 A second major concern reflects these authors' lack of access to the full set of the qualitative data from these studies. They cannot take the original data to arrive at overarching themes similar to a meta-analysis	Meta-ethnography differs from a meta- analysis. We have outlined the steps taken as per Noblit and Hare which does not necessitate going back to the original data. It is the original authors'

of quantitative data from multiple studies. Thus, it is unclear how the themes from these studies lead to the five themes that were identified.	findings as 2nd order concepts which are then synthesised to produce 3rd order concepts.
R1-5 The figure that tries show how these themes relate to each other is simplistic. The paper would benefit from a table summarizing key themes identified by the authors of each of the relevant studies. Thus the reader could example more information from these studies. For example, several studies including Frank et al and Simmonds et al focus on the lack of understanding of nonpharmacologic approaches to managing chronic pain. This is an important, clinically relevant focus that is lost in the theme related to understanding opioids.	In light of your comments, we have re-examined our model. We feel it reflects our interpretation of the data and to us it is accessible rather than overly simplistic, and does describe our data. In this work we have drawn out the third order concepts relevant to our research question; specifically; 'what are peoples' experiences are of both using opioids for CNMP and their attempts to stop taking them' For the majority of our included papers our area of interest was not coterminous with that of the original authors. If we tabulated the themes identified by the authors of the included papers will add a substantial amount of redundant data and detract from the clarity of the analysis. We have in tables 2 and 3 clearly identified how all of the included papers have contributed to our analysis which provides the audit trail needed to demonstrate a robust and reproducible line of argument.
R1-6 Another theme that really isn't elicited relates to patient challenges with accessing opioids - this goes beyond tapering and withdrawal but reflects difficulties patients face with receiving refills of opioid medications on time due to a cumbersome healthcare system, poor communication, and multiple physicians. This was a clear message of several of these studies including Simmonds.	Thank you for this comment. The healthcare system we feel is covered within the Therapeutic alliance theme.
R1-7 The discussion has relatively few references and mostly reflects opinion. Some statements are not based in fact at least in regard to the US experience. For example, on page 29 lines 31 through 33 the authors state that patients are being given opioids as a last resort. This has not been the case in the United States and it is a major reason for the opioid epidemic.	Thank you for this thought provoking comment. We have looked again at this section. . We acknowledge the opioid epidemic especially in the US and have added to the limitations to take this into account more fully. Some papers discuss using opioids as a last resort, although the opioid epidemic, especially in the US suggests they are not always given as a last resort.

R1-8 Another sentence that doesn't make sense is page 29 lines 51 through 52 on where it states that individuals who were concerned about addiction thought it was still reasonable to receive opioids for short-term pain relief. Most patients with chronic pain are looking to use these drugs long term and not for short term pain relief.	Thank you for pointing this out. Our data suggests that patients spoke of trying anything to cope with the pain at that time. The term short term is misleading and we have removed it.
R1-9 Another concern is on page 31 lines 20 through 24 where the term opioid contracts is used. This is unacceptable language because these are not legal contracts and many papers have been written about their being agreements at best. If opioid agreements are addressed, the authors should reference ample literature regarding benefits and challenges.	Thank you for bringing our attention to the terminology we have used. Two of the studies used the term opioid contracts and opioid agreements are also mentioned we have added some text to explain the terminology being used. We have also added a reference. Opioid contracts in some areas of the USA and Canada can make patients feel stigmatised and judged, this effect can be moderated by a good therapeutic relationship, and reframing these as agreements rather than contracts. Some physicians may view contracts/agreements as necessary to guard against uncontrolled dose escalation, repeated demands for replacement of lost or misplaced medication, subversion and illicit opioid intake.
Minor comments R1-10 CASP needs to be spelled out in the abstract Table 1 offers a direct quote of the aims of the included studies which seems superfluous.	We have amended the CASP abbreviation. After considering the aims column we have decided to keep it in as it helps to give readers some further context of how and why the data was obtained.
R1-11 The comment on page 4 about the expertise of the team is not needed	We have included the expertise of the team because it helps to position the research and any potential biases. It is also recommended by CASP see question 6
Reviewer: 2 This is a very well written paper documenting the experiences of patients receiving opioid pain medications for chronic non-cancer pain patients. The authors did a great job synthesizing the literature into emergent themes that describe the challenges faced and support available for these patients. I have a	Thank you for your comments. We are delighted you felt the paper was well written. Thank you.

few concerns, however, I think the authors may be able to clarify them.	
R2-1. It seems that there were about 41% duplicates - is this really true? Why was this? Was this because different systems were pulling the same papers?	We have checked our records and that is correct, different database systems were pulling the same papers.
R2-2. It is unclear which different criteria were used to first exclude 865 records during identification process and then another 1976 during the screening process, please clarify in the manuscript and the PRISMA diagram.	Thank you for highlighting this we have added some text in the flowchart to clarify this. Electronic search terms were used to identify records which were not eligible e.g. paediatric
R2-3. In the PRISMA flow diagram, in the last box of included - please say "meta-ethnography' instead of meta-analysis.	Thank you for pointing this out we have amended.
R2-4. In the eligibility exclusion (n=122), the 3 "not chronic pain" could also go under the "off topic."	Thank you we have considered this and agree. We have changed the flow chart.
R2-5. Table 2 row 1 column 3, Add the six studies with concerns here and then say the remaining 7 have no concerns.	Thank you for highlighting this. We have checked our data and amended this table
R2-6. Table 2 row 1 column 4, what about the 13th study? Same goes for row 2, columns 3 and 4.	a/a
R2-7. Table 2 row 3 column 4, what about the other 3 studies?	a/a
R2-8. Table 2 row 4 column 3, what about the other 5 studies?	a/a

R2-9. Table 2 row 4 column 4, what about the other 2 studies?	a/a
R2-10. Table 2 row 5 column 2, say "(6 studies)" here 11.	a/a
R2-11. Instead of using author initials in the methods, use person neutral language in the main manuscript. All author contributions can be listed in the acknowledgments. E.g., Step 3 could be rewritten as, "Three researchers read the studies such that each study was read at least once and all researchers extracted the second order concepts independently. All three researchers met to discuss and reach"	We have looked at other recent meta-ethnographies and most have initials given in the text which we feel promotes transparency. We therefore regard this as a house style decision to keep them in.
Reviewer: 3 The authors present a meta-ethnography summarizing and integrating qualitative studies about the experience of patients on opioid medication and/or coming off them. This current and timely analysis addresses the important perspective of chronic pain patients in the current discussion on the benefit and risks of chronic opioid treatment. The central theme of constantly balancing was concluded from the studies. The reviewer is not an expert on conducting meta-ethnography, but the methodology seems sound, thorough and exhaustive, and is described in detail (Noblit and Hare's 7 stages, grade-CERQual assessment, eMERGe meta-ethnography reporting) and presented accordingly. The study is well-written with clear conclusions. There are no major comments, but the authors may consider the following thoughts.	Thank you for your positive comments
R3-1 The five themes evolving out of this analysis make sense from a clinical perspective and reflect the general ambivalence of both patients and prescribers. The phrasing, however, of the theme	We have reconsidered this but would like to keep the phrasing as patients discussed gaining knowledge of both good and bad aspects of opioids and we feel the phrasing describes this duality.

"Understanding Opioids: the good and the bad" seems not ideal, since the various and complicated processes of gaining knowledge is the focus and not the realisation of potential benefits, like functional or other improvements. It is interesting, that what now is the focus of most current guidelines, namely to ensure improvement in physical and social functions with opioid appears not as a relevant "good" aspect, except in the context of "the reluctant user".	
R3-2. The authors include 31 studies with the vast majority coming from the US prior to current guidelines and the realisation of the impact of the opioid crisis. They rightly comment on this as a relevant limitation. However, as this period is commonly characterized as a period of uncritical prescribing of opioids, their findings seem somewhat counterintuitive by already highlighting ambivalence and risk. May be there also is an influence (as an additional limitation) of the current justifiably critical perception of opioid treatment on the process of developing the overarching concepts?	Thank you for raising this. We have attempted to minimise bias in our findings by staying true to a meta-ethnographic process. However we do realise that our 3rd order concepts are our interpretation of the data. The authors will have been exposed to news about the opioid crisis however the data, reflected in our themes, showed that there were different experiences in each theme i.e. Not everyone considered opioid use negatively. We have re-read our results and discussion in light of your comments and have added a sentence into the limitation section of the discussion. We acknowledge that interpretation of the data might have been influenced by the current, much more critical perception of opioid use for chronic non-malignant pain.
R3-3. Although the theoretical construct behind this meta-ethnography relies on qualitative analysis of content, background, treatment setting and other patient related factors play a relevant role in shaping this content. A lot of this Information is already summarized in table 1. May be some additional information on the patients could be added as a seperate column, similar to comments like methadone specific or USA Veteran. Or to highlight if none of this Information is available.	We've looked at the column 'comments' in table 1 and decided these were more for the team as they developed this meta-ethnography and have now decided to remove it and have added morphine equivalent data.

VERSION 2 – REVIEW

REVIEWER	Barbara J Turner University of Texas Health Science Center San Antonio
-----------------	---

REVIEW RETURNED	08-Nov-2019
-------------

GENERAL COMMENTS	1) How can a study in which respondents wrote notes in the margins of a document (ref 25) not have any methodological concerns? (Table 2) I am concerned about the subjective nature of these assessments – at least needs to be highlighted in the limitations 2) Would rephrase the comment about opioid prescribing in the US: “Some papers discuss using opioids as a last resort, although the opioid epidemic, especially in the US indicates that the threshold for prescribing opioids was low until recent initiatives to discourage prescribing long-term opioids for chronic pain” – might add a reference to this re initiatives to address the opioid epidemic 3) The limitations are too terse – you should acknowledge that you are not looking (by definition) at primary data rather taking original authors’ findings as 2nd order concepts and synthesizing them to produce 3rd order concepts. Therefore more subjective. You should also acknowledge that most of the studies do not report the dose of opioids that participants were taking, so can’t determine whether they were taking risky doses (>90 MME). 4) Minor – to be more explicit add pediatric studies (age <18) as exclusion (Box 1) 5) The conclusions of the abstract does not really capture your themes well – would rephrase.
---

REVIEWER	Shabbar I Ranapurwala University of North Carolina at Chapel Hill, Department of Epidemiology
-----------------	--

REVIEW RETURNED	23-Nov-2019
-------------

GENERAL COMMENTS	The authors have addressed all of my concerns. I have no outstanding concerns.
--

VERSION 2 – AUTHOR RESPONSE

Reviewer 1	
Many thanks for your comments we have responded to each point below	
1) How can a study in which respondents wrote notes in the margins of a document (ref 25) not have any methodological concerns? (Table 2) I am concerned about the subjective nature of these assessments – at least needs to be highlighted in the limitations	Thank you. We have added a comment in table 2. *That comments were written in the margin is a potential limitation.
2) Would rephrase the comment about opioid prescribing in the US: “Some papers discuss using opioids as a last resort, although the opioid epidemic, especially in the US indicates that the threshold for prescribing opioids was low until recent initiatives to discourage prescribing long-term opioids for chronic pain” – might add a reference to this re initiatives to address the opioid epidemic	Thank you we have added this sentence with a reference. Schieber et al 2019

3) The limitations are too terse – you should acknowledge that you are not looking (by definition) at primary data rather taking original authors' findings as 2nd order concepts and synthesizing them to produce 3rd order concepts. Therefore more subjective. You should also acknowledge that most of the studies do not report the dose of opioids that participants were taking, so can't determine whether they were taking risky doses (>90 MME).	We have explained the 2nd and 3rd order concepts in the methods, that this is a recognised meta-ethnographic method and have added that this is the reviewers' interpretation of second order concepts. We have added in the limitations section. Not all studies gave morphine equivalent data so we cannot determine what proportion were taking high, medium or low doses.
4) Minor – to be more explicit add pediatric studies (age <18) as exclusion (Box 1)	We agree and have added
5) The conclusions of the abstract does not really capture your themes well – would rephrase.	We have rephrased this section to reflect the themes more closely. People taking opioids were constantly balancing tensions, not always wanting to take opioids, weighing the pros and cons of opioids but feeling they had no choice because of the pain. They frequently felt stigmatised, were not always 'on the same page' as their health care professional and changes in opioid use were often challenging.